Creatine kinase mitochondrial 2 promotes the growth and progression of colorectal cancer via enhancing Warburg effect through lactate dehydrogenase B

Cai Shasha 1
Xia Qingqing 1
Duan Darong 1
Fu Junhui 2
Wu Zhenxing 3
Yang Zaixing 1 yangzaixingdiyi@163.com
Yu Changfa 1 yuchangfa123@163.com
1 Laboratory Medicine, Taizhou First People’s Hospital, Huangyan Hospital of Wenzhou Medical University , Taizhou , China
2 General Surgery, Taizhou First People’s Hospital, Huangyan Hospital of Wenzhou Medical University , Taizhou , China
3 Gastroenterology, Taizhou First People’s Hospital, Huangyan Hospital of Wenzhou Medical University , Taizhou , China
Ozdag Sevgili Hilal
Electronic publication date: 2024 Jun 28
Publication date: 2024
Volume: 12
Electronic Location ID: e17672
Received 2024 Jan 15; Accepted 2024 Jun 12
Copyright: © 2024 Cai et al.
Copyright year: 2024
Copyright holder: Cai et al.
License: This is an open access article distributed under the terms of the Creative Commons Attribution License, which permits unrestricted use, distribution, reproduction and adaptation in any medium and for any purpose provided that it is properly attributed. For attribution, the original author(s), title, publication source (PeerJ) and either DOI or URL of the article must be cited.
License URL: https://creativecommons.org/licenses/by/4.0/

Keywords: Creatine kinase mitochondrial 2, Colorectal cancer, Warburg effect, Lactate dehydrogenase B

Funding: Zhejiang Provincial Natural Science Foundation of China LQ21H200003 Medical and Health Technology Project in Zhejiang Province 2020KY1045 Taizhou Science and Technology Bureau Project in Zhejiang Province 22ywb43, 22ywa30, 23ywa28 This research was supported by the Zhejiang Provincial Natural Science Foundation of China under Grant No. LQ21H200003, the Medical and Health Technology Project in Zhejiang Province (2020KY1045), and the Taizhou Science and Technology Bureau Project in Zhejiang Province (22ywb43, 22ywa30, 23ywa28). The funders had no role in study design, data collection and analysis, decision to publish, or preparation of the manuscript.

==============================
Background

Mitochondrial creatine kinase (MtCK) plays a pivotal role in cellular energy metabolism, exhibiting enhanced expression in various tumors, including colorectal cancer (CRC). Creatine kinase mitochondrial 2 (CKMT2) is a subtype of MtCK; however, its clinical significance, biological functions, and underlying molecular mechanisms in CRC remain elusive.

Methods

We employed immunohistochemical staining to discern the expression of CKMT2 in CRC and adjacent nontumor tissues of patients. The correlation between CKMT2 levels and clinical pathological factors was assessed. Additionally, we evaluated the association between CKMT2 and the prognosis of CRC patients using Kaplan-Meier survival curves and Cox regression analysis. Meanwhile, quantitative reverse transcription polymerase chain reaction (qRT-PCR) was used to detect the expression levels of CKMT2 in different CRC cell lines. Finally, we explored the biological functions and potential molecular mechanisms of CKMT2 in CRC cells through various techniques, including qRT-PCR, cell culture, cell transfection, western blot, Transwell chamber assays, flow cytometry, and co-immunoprecipitation.

Results

We found that CKMT2 was significantly overexpressed in CRC tissues compared with adjacent nontumor tissues. The expression of CKMT2 is correlated with pathological types, tumor size, distant metastasis, and survival in CRC patients. Importantly, CKMT2 emerged as an independent prognostic factor through Cox regression analysis. Experimental downregulation of CKMT2 expression in CRC cell lines inhibited the migration and promoted apoptosis of these cells. Furthermore, we identified a novel role for CKMT2 in promoting aerobic glycolysis in CRC cells through interaction with lactate dehydrogenase B (LDHB).

Conclusion

In this study, we found the elevated expression of CKMT2 in CRC, and it was a robust prognostic indicator in CRC patients. CKMT2 regulates glucose metabolism via amplifying the Warburg effect through interaction with LDHB, which promotes the growth and progression of CRC. These insights unveil a novel regulatory mechanism by which CKMT2 influences CRC and provide promising targets for future CRC therapeutic interventions.

Introduction

Colorectal cancer (CRC) stands as the most prevalent malignant tumor in the digestive system worldwide, ranking third in incidence and second in mortality among all malignancies (Morgan et al., 2023). In recent years, the incidence and mortality rates of CRC increase rapidly, notably in developing countries, including China (Biller & Schrag, 2021). CRC poses a significant threat to human health and places a substantial medical burden on society. Despite advancements in diagnostic techniques and treatment modalities for CRC, there has been limited improvement in the early-stage diagnosis rate and the mortality rate for late-stage patients undergoing radiotherapy and chemotherapy. Progress in effective, noninvasive diagnostic markers and treatment strategies is imperative to reduce CRC mortality and enhance the quality of life for patients.

Aerobic glycolysis (also known as the Warburg effect) is regarded as the major energy production mechanism in cancers. In addition, oxidative phosphorylation and creatine shuttle are also known to be responsible for energy metabolism in cancers. Metabolic reprogramming is a distinct hallmark of cancer. Hence, enzymes and proteins related to glucometabolic may play an important role in tumor generation. Tumors are classified into four metabolic subtypes based on genomic and transcriptomic alterations, finding that the glycolytic subtype has the worst prognosis compared to other subgroups (Zhang et al., 2021). Lactate dehydrogenase (LDH), one of the key enzymes in glycolysis, is encoded by the lactate dehydrogenase A and B (LDHA and LDHB) genes. It uses NADH/NAD+ as a cosubstrate to catalyze the interconversion of pyruvate and lactate. Only the double knockout of LDHA and LDHB can inhibit LDH activity and lactate secretion. Transmission electron microscopy (TEM) shows that LDHB is located in the inner membrane of mitochondria, playing a significant role in glycolysis and mitochondrial metabolism (Chen et al., 2016).

Creatine kinase mitochondrial 2 (CKMT2), also known as sarcomeric mitochondrial creatine kinase (sMtCK), demonstrates tissue specificity and exclusive expression in skeletal muscles. CKMT2 is closely related with the output of adenosine triphosphate (ATP) through adenine nucleotide translocase or carriers, serving as a crucial participant in ATP synthesis and respiratory chain activity. High expression of CKMT2 can enhance cellular vitality and mitochondrial ATP levels, improve mitochondrial function, protect mitochondrial membrane potential, and reduce the generation of reactive oxygen species (ROS) by mediating the proliferator-activated receptor γ coactivator-1α/estrogen-related receptor-α (PGC-1α/ERRα) pathway (Park et al., 2021).

It belongs to the mitochondrial creatine kinase (MtCK) family, alongside ubiquitous MtCK (uMtCK or CKMT1), expressed in various tissues such as smooth muscle, brain, and kidney (Kita et al., 2023). MtCK is situated in the intermembrane space between the inner and outer mitochondrial membranes, playing an essential role in connecting mitochondria, sarcoplasmic reticulum, myofibers, and the cell nucleus within the intricate network of metabolic energy transfer. It facilitates mitochondrial energy supply via the phosphocreatine shuttle. Abnormal upregulation of uMtCK has been observed in malignancies, including non-small-cell lung cancer (Yang et al., 2023), nasopharyngeal carcinoma (Lan et al., 2019), gastric cancer (Mi et al., 2023), and CRC (Tang et al., 2022). Recently, a potential association between CKMT2 and metabolic dysregulation in gastric cancer has also been suggested (Wen et al., 2020). Creatine kinase B (CKB) and creatine kinase M (CKM), which are two isoforms of cytosolic creatine kinase isoenzymes, have been reported to be associated with various glycolytic enzymes. They promote the conversion of ATP produced by glycolysis into phosphocreatine (PCr), thus maintaining ATP homeostasis by decoupling glycolysis (Yan, 2016). The suppression of CKB expression inhibits glycolysis in ovarian cancer cells, reduces glucose consumption and lactate concentration, and increases ROS and oxygen consumption. Consequently, this suppresses ovarian cancer cell proliferation and induces apoptosis (Li et al., 2013). Meanwhile, the relationships between MtCK and glycolysis were also reported in the previous literatures. A study indicated that CKMT1 was positively associated with HK2 expression, and may strengthen aerobic glycolysis in GC cells (Mi et al., 2023). Chen et al. (2022) classified CRC into different metabolic subgroups and found high expression of CKMT2 in patients with the glycolytic subtype, suggesting that CKMT2 may play a regulatory role in cancer cell proliferation and tumorigenesis through the glycolytic pathway. This research evaluates the role of CKMT2 in the occurrence and development of CRC and its underlying mechanisms, and it uncovers a novel biomarker for CRC that could be used to facilitate the development of new diagnosis and treatment strategies for CRC.

Materials and Methods

Study population

This is a prospective study. We enrolled 220 newly diagnosed patients with CRC from June 2020 to December 2022 at the First People’s Hospital of Taizhou City, Zhejiang Province, China. None of the patients had undergone surgical, radiotherapy, or chemotherapy treatments before they enrolled, and all the patients exhibited a clear pathological diagnosis. Patients with metastatic tumors and multi-organ tumors were excluded. All participants were subjected to follow-up assessments, with the endpoint set at the patient’s death or September 2023. This study was approved by the ethical committee of the First People’s Hospital of Taizhou City, and all patients signed the informed consent form before enrollment (Ethical Application Ref: 2019-KY-009-03).

Immunohistochemistry

The expression levels of CKMT2 and LDHB in CRC tissues were tested via immunohistochemistry. Results were determined by multiplying the percentage of positive cells (0–100%) and the positive intensity (0 indicated no staining, one indicated weak staining, two indicated moderate staining, and three indicated strong staining). The results were judged by two pathologists independently, if the results were not matched, a third pathologist will make a final decision. Antibodies for CKMT2 and LDHB were obtained from Abcam, Cambridge, UK (ab227440, ab53292), and the immunohistochemistry reagent kit was purchased from Beijing Solarbio Science & Technology Co., Ltd., Beijing, China. Experiments were performed according to the instruction from the kit.

Quantitative reverse transcription polymerase chain reaction (qRT-PCR)

The expression profile of CKMT2 across human colonic epithelial cells (NCM460) and various human CRC cell lines (SW480, DLD-1, SW620, HCT116, HCT15) was evaluated by qRT-PCR. The CKMT2 expressions in SW480 and NCM460 cells after transfection with CKMT2 siRNA were also detected by qRT-PCR. Total RNA was extracted using the TRIzol method (TaKaRa Bio Inc., Shiga, Japan), and the PrimeScriptTM RT reagent kit (RR036A; TaKaRa, Shiga, Japan) was used to synthesize the first strand of cDNA. The reaction system comprised 2 ug of total RNA and five times of reverse transcription reagent. Amplification was subsequently performed with the cDNA as a template, using the TB GreenTM Premix Ex TaqTM kit (RR420A; TaKaRa, Shiga, Japan) on the ABI 7500 Real-Time PCR system. CKMT2 mRNA expressions in different samples were analyzed with the formula 2−ΔΔCt, and normalized to glyceraldehyde 3-phosphate dehydrogenase (GAPDH) as the internal reference. All samples were measured in triplicates.

Cell culture

Human colonic epithelial cell line (NCM460) and human CRC cell lines (SW480, DLD-1, SW620, HCT116, and HCT15) were obtained from FuHeng Biology (Shanghai, China). Cells were cultured at 37 °C with 5% CO2 in Dulbecco’s modified Eagle medium (DMEM) containing 10% fetal bovine serum (Gibco, Thermo Fisher Scientific, MA, USA) and 1% (v/v) penicillin/streptomycin (Gibco, Thermo Fisher Scientific, MA, USA). NCM460 and SW480 cells were cultured in DMEM (2317091; Viva Cell Biosciences, Shanghai, China), whereas DLD-1, SW620, HCT116, HCT15 cells were cultured in RPMI-1640 (2317095; Viva Cell Biosciences, Shanghai, China). All cells were harvested in the logarithmic growth phase. All cell experiments were repeated three times.

Cell transfection

CKMT2 siRNA (sense 5′-GCAACAAGGUGACACCCAATT-3′, antisense 5′-UUGGGUGUCACCUUGUUGCTT-3′) and its negative control (NC) siRNA (sense 5′-UUCUCCGAACGUGUCACGUTT-3′, antisense 5′-ACGUGACACGUUCGGAGAATT-3′) were purchased from Sigma (Shanghai, China). SW480 cells were transfected with siRNA using Lipofectamine 3000 (L3000001; Thermo Fisher Scientific, MA, USA) and Opti-MEM (31985070; Gibco, MA, USA). Following 48 h of transfection, the transfection efficiency was verified using qRT-PCR and western blotting.

Cell migration

SW480 cells were seeded in the upper chamber of a 24-well Transwell plate. Upon reaching 30–50% confluence, they were transfected with CKMT2 siRNA or NC siRNA. After 48 h of transfection, the upper chambers were removed, and a cotton swab was used to eliminate non-migrated cells. Subsequently, the cells were fixed with 4% paraformaldehyde for 15 min, washed three times with phosphate-buffered saline (PBS), stained with 0.5% crystal violet solution for 10 min, rinsed with PBS, and then photographed under a microscope (Olympus).

Cell apoptosis

The apoptosis of SW480 cells were tested after 48 h of transfection with CKMT2 siRNA or NC siRNA. The Annexin V-FITC Apoptosis Detection Kit (CA1020, Solarbio, Beijing, China) was used. A minimum of 104 cells per sample were evaluated using flow cytometry (DxFLEX; Beckman Coulter, Suzhou, Jiangsu, China).

Western blotting

SW480 cells were lysed using an appropriate volume of radioimmunoprecipitation assay buffer (Beyotime Biotechnology, Shanghai, China) and sonicated in an ice-water bath for 10 mins. After centrifugation at 12,000 rpm at 4 °C for 10 mins, the supernatant was collected. Protein concentration was measured and adjusted to 1 µg/µL. For electrophoresis, we prepared a 10% sodium dodecyl-sulfate polyacrylamide gel electrophoresis (SDS-PAGE) gel, loading each well with 20 µg of protein. The proteins were subsequently transferred to a polyvinylidene fluoride membrane, blocked with 5% skimmed milk at room temperature for 1 h. Following TBST washing, the membrane was incubated with the respective primary antibodies at 4 °C overnight. The membrane was subsequently washed three times with TBST for 5 mins each time and was then incubated at room temperature for 1 h with horseradish peroxidase-conjugated anti-rabbit (A0208; Beyotime, Shanghai, China) or anti-mouse (A0192; Beyotime, Shanghai, China) secondary antibodies. After being washed by TBST for three times, the membrane was developed using an enhanced chemiluminescence substrate (Beyotime, Shanghai, China, P0018M) and exposed for photographic documentation. Antibodies for CKMT2 and GAPDH were sourced from HUABIO (ER64987, ET1601-4), and the LDHB antibody was purchased from Proteintech (66425-1-lg).

Co-immunoprecipitation (Co-IP)

Following the designated treatments, SW480 cells were lysed with Co-IP cell lysis buffer (R0030; Solarbio, Beijing, China), supplemented with 1 mM phenylmethylsulfonyl fluoride (ST506; Beyotime, Shanghai, China) and 1% protein phosphatase inhibitor cocktail (P1260; Solarbio, Beijing, China) for 30 mins on ice. The cell lysate was then centrifuged, and the supernatant was collected for further analysis. For input examination, 20 µL of the supernatant was reserved. The remaining was mixed with 30 µL of protein A/G CoIP magnetic beads (SB-AG001; Share-Bio, Shanghai, China) and rotated at 4 °C for 30 mins. Subsequently, the supernatant was collected, incubated with rabbit anti-CKMT2 (13207-1-AP; Proteintech, Rosemont, IL, USA), or mouse anti-LDHB (Proteintech, Rosemont, IL, 66425-1-lg), and subjected to rotation at 4 °C overnight. Following this, the mixture was combined with 30 µL of protein A/G CoIP magnetic beads and rotated at 4 °C for 4 h. The beads were collated in a magnetic test-tube rack, washed three times using lysis buffer, and eluted with 1 × SDS-loading buffer. The supernatant was collected, heated, and then subjected to western blotting.

Statistical analyses

SPSS 22.0 software (SPSS, Inc., Chicago, IL, USA) and GraphPad Prism 8 (GraphPad Software Inc., San Diego, CA, USA) were used for the data analysis. The Student’s t-test was employed for comparing two groups of continuous variables following a normal distribution, whereas analysis of variance (ANOVA) was used for multiple group comparisons. We employed the Mann-Whitney U test for non-normally distributed variables. Kaplan-Meier survival curves were plotted, and significance was determined using the log rank test. Univariate and multivariate Cox regression analyses were conducted to identify prognostic factors for CRC patients. Statistical significance was determined when p-value < 0.05.

Results

CKMT2 is highly expressed in CRC tissues and cell lines

A comprehensive immunohistochemical analysis was conducted on CRC tissues and their corresponding distant nontumor counterparts sourced from a cohort of 220 patients with CRC. The staining intensity of CKMT2 in cancer tissues exhibited a robust pattern, contrasting with the minimal to absent staining observed in adjacent nontumor tissues (Figs. 1A–1D). Notably, the immunohistochemical scoring indicated a marked elevation in the expression of CKMT2 in cancer tissues (1.60 ± 0.53) compared to adjacent nontumor tissues (0.45 ± 0.29), demonstrating statistical significance (t = 28.397, p < 0.001) (Fig. 1E). Using qRT-PCR, the expression profile of CKMT2 was evaluated across human colonic epithelial cells (NCM460) and various human CRC cell lines (SW480, DLD-1, SW620, HCT116, HCT15), which are widely used in vitro cancer research. The results elucidated a significant increase in CKMT2 expression in SW480 and SW620 cells when compared with NCM460 (p < 0.05) (Fig. 1F). However, a distinct decrease in the expression of CKMT2 in DLD-1 and HCT15 cells compared to NCM460 was also observed (p < 0.01) (Fig. 1F). The expression of CKMT2 in HCT116 cells showed an increasing trend, though statistically not significant. For further validation, an analysis of The Cancer Genome Atlas (TCGA) database, a widely acknowledged repository of cancer gene information, revealed a significant upregulation of CKMT2 mRNA in CRC tissues in contrast to normal controls (p < 0.001) (Fig. 1G).

Figure 1 Expression of CKMT2 in colorectal cancer (CRC).

(A) CKMT2 expression in distal para-cancerous tissues of patients with CRC (×100). (B) CKMT2 expression in distal para-cancerous tissues of patients with CRC (×400). (C) CKMT2 expression in CRC tissues (×100). (D) CKMT2 expression in CRC tissues (×400). (E) Immunohistochemical scoring of CKMT2 in tissues from 220 patients with CRC. (F) Expression of CKMT2 in human colon epithelial cells and human CRC cells. (G) Analysis from TCGA database showing significantly increased expression of CKMT2 in CRC (625 cases, 51 controls) (p < 0.001). *p < 0.05, **p < 0.01, ***p < 0.001.

CKMT2 expression is related with some clinical pathological factors in CRC patients

After collecting clinical information from 220 patients with CRC, an in-depth analysis was conducted to discern the relationship between CKMT2 expression and various clinical pathological factors (Table 1). CKMT2 expression level is higher in patients with adenocarcinoma patients compared with mucinous carcinoma (t = 2.169, p = 0.031). Furthermore, patients with a tumor size ≥5 cm exhibited enhanced CKMT2 expression compared to their counterparts with a tumor size <5 cm (t = 2.294, p = 0.023). Patients with distant metastasis showed noteworthy results, presenting higher CKMT2 expression than those without metastasis (t = 2.304, p = 0.022). Furthermore, in survival patients, the expression level of CKMT2 was lower than those who were not alive (t = 2.163, p = 0.032). There were no statistically significant correlations between CKMT2 expression and patient gender, age, smoking status, alcohol consumption, tumor differentiation degree, TNM stage, tumor infiltration depth, lymph node metastasis, or carcinoembryonic antigen (CEA) levels (p > 0.05).

Table 1 Relationship between CKMT2 immunohistochemical score in cancer tissues and clinical pathological features in 220 patients with colorectal cancer.

Clinical pathological features	N	CKMT2	t/F value	p value	
Gender			t = 0.690	0.491	
Male	137	1.62 ± 0.49			
Female	83	1.57 ± 0.59			
Age (year)			t = 0.194	0.847	
<70	116	1.60 ± 0.46			
≥70	104	1.59 ± 0.59			
Tumor locationa			t = 1.024	0.307	
Colon	132	1.63 ± 0.53			
Rectum	84	1.56 ± 0.53			
Smoking			t = 0.136	0.892	
Yes	61	1.59 ± 0.47			
NO	159	1.60 ± 0.55			
Alcohol drinking			t = 0.142	0.887	
Yes	52	1.59 ± 0.49			
No	168	1.60 ± 0.54			
Pathological typea			t = 2.169	0.031	
Adenocarcinoma	194	1.61 ± 0.52			
Mucinous carcinoma	17	1.33 ± 0.54			
Tumor differentiation degree			F = 0.559	0.573	
High differentiated	21	1.48 ± 0.47			
Moderately differentiated	164	1.61 ± 0.54			
low differentiated	35	1.61 ± 0.51			
Tumor size (cm)			t = 2.294	0.023	
≥5	92	1.69 ± 0.53			
<5	128	1.53 ± 0.52			
TNM stage			F = 1.661	0.176	
Stage I	35	1.49 ± 0.45			
Stage II	82	1.69 ± 0.51			
Stage III	81	1.54 ± 0.56			
Stage IV	22	1.62 ± 0.58			
Tumor infiltration depth			F = 1.162	0.325	
T1	10	1.44 ± 0.58			
T2	33	1.49 ± 0.43			
T3	45	1.57 ± 0.53			
T4	132	1.65 ± 0.54			
Lymphatic metastasis			t = 1.580	0.115	
Yes	92	1.53 ± 0.56			
No	128	1.64 ± 0.50			
Distant metastasis b			t = 2.304	0.022	
Yes	49	1.75 ± 0.53			
No	171	1.55 ± 0.52			
CEA (ng/mL)			t = 0.688	0.492	
>5	97	1.62 ± 0.54			
≤5	123	1.57 ± 0.52			
Survival outcomeb			t = 2.163	0.032	
Death	53	1.73 ± 0.57			
Survival	167	1.55 ± 0.51			
Notes:

a Data not included in the statistics were a mixture of two classifications.

b The assessment cutoff time was September 2023.

CKMT2 expression is related with the prognosis of CRC patients

A total of 220 enrolled CRC patients received follow-up, outcome information, including metastasis and mortality were collected. Utilizing patient survival as the state variable and CKMT2 expression results as the test variable, a receiver operating characteristic (ROC) curve was plotted (Fig. 2A), revealing an area under the curve of 0.602 (p = 0.025). The optimal cutoff value was determined to be 1.85. Patients were stratified based on both the cutoff value (1.85) and median (1.60), and Kaplan-Meier survival curves were generated, with the endpoint being the occurrence of distant metastasis or death. In Fig. 2B, when CKMT2 was categorized into high-expression (CKMT2 > 1.85) and low-expression (CKMT2 ≤ 1.85) groups using the cutoff value, the average survival time for the low-expression group was 34.32 months (95% confidence interval [CI] [32.72–35.92]), whereas that for the high-expression group averaged 27.59 months (95% CI [24.95–30.22]). Patients with high CKMT2 expression exhibited significantly lower average survival time and overall survival rates than the low-expression group (log rank p < 0.001). Similarly, when CKMT2 was divided using the median into high-expression (CKMT2 > 1.60) and low-expression (CKMT2 ≤ 1.60) groups, the survival time for the low-expression group averaged 34.09 months (95% CI: [32.24–35.94]), and that for the high-expression group averaged 30.40 months (95% CI [28.13–32.68]). Patients with high CKMT2 expression exhibited significantly lower average survival time and overall survival rates than the low-expression group (log rank p = 0.015) (Fig. 2C). Furthermore, the free from distant metastasis (FDM) survival rate was employed to assess the relationship between CKMT2 expression and distant metastasis in patients with CRC. Results indicated that patients with high CKMT2 expression exhibited significantly lower FDM than those exhibiting low expression (cutoff value grouping, log rank p = 0.007; median grouping, log rank p = 0.031) (Figs. 2D, 2E). Prognostic information of patients with CRC from TCGA was downloaded to analyze the relationship between CKMT2 expression and overall survival time, further confirming the effects of CKMT2 levels on patient survival (Fig. 2F). The results demonstrated that patients with high CKMT2 expression exhibited significantly lower overall survival rates than those with low expression (log rank p = 0.039).

Figure 2 Assessment of CKMT2 expression for the prognosis of patients with CRC.

(A) ROC curve depicting CKMT2 expression and survival outcomes. (B) Grouping based on cutoff values to evaluate the differences in the overall survival between high and low CKMT2 expression groups. (C) Grouping based on the median to evaluate the differences in the overall survival between high and low CKMT2 expression groups. (D) Grouping based on cutoff values to evaluate the differences in FDM survival between high and low CKMT2 expression groups. (E) Grouping based on the median to evaluate the differences in FDM between high and low CKMT2 expression groups. (F) Analysis of the overall survival difference between high and low CKMT2 expression groups in the TCGA database.

CKMT2 is an independent prognostic factor for CRC patients

Univariate and multivariate COX regression analysis were performed to evaluate the prognostic factors for CRC patients. Age (hazard ratio (HR): 3.400, 95% CI [1.865–6.198], p < 0.001), differentiation grade (HR: 2.767, 95% CI [1.627–4.704], p < 0.001), tumor size (HR: 1.921, 95% CI [1.117–3.304], p = 0.018), TNM stage (HR: 1.867, 95% CI [1.332–2.617], p < 0.001), CEA level (HR: 2.967, 95% CI [1.664–5.290], p < 0.001), and CKMT2 level (HR: 2.925, 95% CI [1.686–5.076], p < 0.001) were found to be associated with the prognosis of patients with CRC (Table 2). Some factors, such as gender, tumor location, smoking, alcohol consumption, and pathological type showed no significant correlation with patient prognosis (p > 0.05). Factors with p < 0.01 in the univariate analysis were included in the multivariate Cox regression analysis. The results revealed that CKMT2 level (HR: 1.929, 95% CI [1.086–3.426], p = 0.025) could serve as an independent factor influencing the prognosis in CRC patients. In addition, age (HR: 3.538, 95% CI [1.848–6.773], p < 0.001), TNM stage (HR: 1.856, 95% CI [1.287–2.679], p = 0.001), CEA level (HR: 2.051, 95% CI [1.129–3.727], p = 0.018) shared the same effect.

Table 2 Univariate and multivariate cox regression analysis of prognostic factors in patients with CRC.

Prognostic factors	Univariate analysis	Multivariate analysis	
HR	95% CI	p value	HR	95%CI	p value	
Gender (Female: Male)	1.012	[0.580–1.765]	0.966				
Age (≥70: <70 yr)	3.400	[1.865–6.198]	<0.001	3.538	[1.848–6.773]	<0.001	
Tumor location (Rectum: colon)	0.735	[0.434–1.246]	0.253				
Smoking (Yes: No)	1.153	[0.647–2.054]	0.630				
Alcohol consumption (Yes: No)	1.300	[0.723–2.338]	0.381				
Histological type (mucinous carcinoma: adenocarcinoma)	0.876	[0.436–1.759]	0.710				
Differentiation degree (low: moderately: high)	2.767	[1.627–4.704]	<0.001	1.286	[0.731–2.263]	0.383	
Tumor size (≥5: <5 cm)	1.921	[1.117–3.304]	0.018				
TNM stage (IV: III: II: I)	1.867	[1.332–2.617]	<0.001	1.856	[1.287–2.679]	0.001	
CEA (>5: ≤5 ng/mL)	2.967	[1.664–5.290]	<0.001	2.051	[1.129–3.727]	0.018	
CKMT2 (>1.85: ≤1.85)	2.925	[1.686–5.076]	<0.001	1.929	[1.086–3.426]	0.025	

Inhibition of CKMT2 expression affects migration and apoptosis of CRC cells

Drawing on the distinctive expression patterns of CKMT2 across various CRC cell lines (Fig. 1F), the SW480 cell line was chosen for further exploration. Initially, CKMT2 siRNA was transfected into SW480 and NCM460 cells to establish cell lines with diminished CKMT2 expression. QRT-PCR analysis demonstrated a substantial reduction in CKMT2 mRNA expression in the siCKMT2 group compared to the NC group (p < 0.001) (Figs. 3A, 3B). Subsequent western blotting validated the successful transfection, demonstrating a marked decrease in CKMT2 protein levels following siCKMT2 transfection (Fig. 3C). The effect of CKMT2 gene silencing on cell migration was evaluated using the Transwell chamber assay, revealing a noteworthy inhibition of cell migration activity in the siCKMT2 group compared to the NC group (Fig. 3D), accompanied by a significant decrease in the cell migration rate (t = 7.292, p < 0.001) (Fig. 3E). After a 48-h transfection period, flow cytometry was employed to examine cell apoptosis, revealing a significantly increased apoptosis rate in the siCKMT2 group compared to that in the NC group (t = 43.89, p < 0.001) (Figs. 3F, 3G).

Figure 3 The effect of CKMT2 expression inhibition on CRC cell migration and apoptosis.

(A) qRT-PCR analysis of CKMT2 expression in NCM460 cells after transfection with CKMT2 siRNA. (B) qRT-PCR analysis of CKMT2 expression in SW480 cells after transfection with CKMT2 siRNA. (C) Western blotting of CKMT2 expression in NCM460 cells and SW480 cells after CKMT2 siRNA transfection. (D) Cell migration after CKMT2 silencing. (E) Changes in cell migration after CKMT2 silencing. (F) Flow cytometry assessment of apoptosis after CKMT2 silencing. (G) Changes in apoptosis after CKMT2 silencing. ***p < 0.001.

CKMT2 impacts the glycolytic metabolism in CRC Cells

We identified the proteins have interactions with CKMT2 from the STRING database (https://cn.string-db.org/) (Fig. 4A), functional enrichment analysis of these associated genes was performed via Database for Annotation, Visualization and Integrated Discovery (DAVID) (https://david.ncifcrf.gov/). Gene Ontology (GO) enrichment analysis revealed that CKMT2 and its interacting factors play a pivotal role in pathways related to sugar metabolism, including “lactate metabolic process,” “pyruvate metabolic process,” “ATP biosynthetic process,” and “L-lactate dehydrogenase activity” (Fig. 4B). Kyoto Encyclopedia of Genes and Genomes (KEGG) enrichment analysis highlighted “glycolysis/gluconeogenesis” and “pyruvate metabolism” as key pathways potentially regulated by CKMT2 interacting factors in CRC (Fig. 4C). We performed further experiments to investigate whether CKMT2 regulates sugar metabolism in CRC cells. Notably, CKMT2 inhibition in NCM460 cells did not significantly alter lactate concentration (Fig. 4D) and glucose uptake (Fig. 4E) levels (p > 0.05). However, in SW480 cells with low CKMT2 expression, we observed a significant decrease in lactate concentration (t = 5.096, p = 0.007) (Fig. 4F) and glucose uptake (t = 6.197, p = 0.003) (Fig. 4G).

Figure 4 CKMT2 enhances glucose metabolism in CRC cells.

(A) Functional protein association network analysis of CKMT2 using the String website. (B) GO enrichment analysis of CKMT2 and its associated genes analyzed using DAVID. (C) KEGG enrichment analysis of CKMT2 and its associated genes analyzed using DAVID. (D) Changes in lactate concentration after silencing CKMT2 in NCM460 cells. (E) Changes in glucose uptake after silencing CKMT2 in NCM460 cells. (F) Changes in lactate concentration after silencing CKMT2 in SW480 cells. (G) Changes in glucose uptake after silencing CKMT2 in SW480 cells. **p < 0.01.

CKMT2 promotes the Warburg effect by upregulating LDHB in CRC cells

To evaluate the mechanism underlying the regulation of the Warburg effect by CKMT2, we analyzed the correlation between CKMT2 and glycolysis-related gene expression in colon adenocarcinoma and rectum adenocarcinoma from the Gene Expression Profiling Interactive Analysis (GEPIA) database (http://gepia.cancer-pku.cn/index.html) (Figs. 5A–5M). The results revealed a negative correlation between CKMT2 mRNA expression and HK2 mRNA (p = 0.029, R = −0.11) and PKM mRNA (p = 0.009, R = −0.14) expression. Conversely, a positive correlation was noted with LDHB mRNA (p = 0.0016, R = 0.16) and PFKM mRNA (p = 3.8e−5, R = 0.21). No significant correlations were noted with the expression of glycolysis-related genes, including ALDOA, ALDOB, ENO1, G6PC, LDHA, PGAM1, PGK1, SLC2A1, and SLC2A4 (p > 0.05). Subsequently, qRT-PCR was employed to examine expression changes of HK2, LDHB, PFKM, and PKM genes in SW480 cells after CKMT2 gene silencing. The results demonstrated a significant decrease in LDHB expression (t = 5.966, p = 0.004) and a modest reduction in HK2 expression (t = 4.061, p = 0.015) following CKMT2 inhibition. However, the expression of PFKM and PKM did not significantly change (p > 0.05) (Fig. 5N). The results of qRT-PCR showed distinct decreases both in LDHB and HK2 expressions. However, the GEPIA database revealed a negative correlation between LDHB and HK2 expression. So, HK2 was not employed for subsequent research. Furthermore, western blotting confirmed that CKMT2 silencing led to the significant downregulation of LDHB expression in SW480 cells (Fig. 5O). Combining the correlation analysis from the GEPIA database and the relative expression changes of glycolysis-related enzymes in SW480 cells, we propose that CKMT2 promotes the Warburg effect by upregulating LDHB expression in CRC cells.

Figure 5 Analysis of CKMT2 expression and glycolysis-related enzymes.

(A–M) Analysis of the correlation between CKMT2 and the expression of ALDOA (A), ALDOB (B), ENO1 (C), G6PC (D), HK2 (E), LDHA (F), LDHB (G), PFKM (H), PGAM1 (I), PGK1 (J), PKM (K), SLC2A1 (L), and SLC2A4 (M) based on the GEPIA database. (N) Relative changes in the expression of HK2, LDHB, PFKM, and PKM detected using qRT-PCR after the inhibition of CKMT2. (O) Relative changes in the expression of LDHB detected using western blotting after the inhibition of CKMT2. *p < 0.05, **p < 0.01.

Additionally, we substantiated the correlation between CKMT2 and LDHB in CRC tissues. Immunohistochemical staining of CKMT2 and LDHB in CRC tissues and distal para-cancerous tissues is presented in Figs. 6A–6H. The assessment of LDHB expression in 30 CRC tissue samples revealed a significantly higher expression in CRC tissues compared to adjacent normal tissues, demonstrating a statistically significant difference (t = 10.77, p < 0.001) (Fig. 6I). Correlation analysis unveiled a positive correlation between CKMT2 and LDHB expression in CRC tissues (r = 0.374, p = 0.042) (Fig. 6J). To further confirm the association between CKMT2 and LDHB, Co-IP was performed. The results showed that in SW480 cells, LDHB protein was detectable in the immunoprecipitated products of CKMT2, and vice versa. Notably, both CKMT2 and LDHB proteins were not detected in the immunoprecipitated products with IgG, as depicted in Fig. 6K. Thus, the interaction between CKMT2 and LDHB was detected.

Figure 6 CKMT2 and LDHB correlation.

(A) CKMT2 expression in distal para-cancerous tissues of patients with CRC (×100). (B) CKMT2 expression in distal para-cancerous tissues of patients with CRC (×400). (C) CKMT2 expression in CRC tissues (×100). (D) CKMT2 expression in CRC tissues (×400). (E) LDHB expression in distal para-cancerous tissues of patients with CRC (×100). (F) LDHB expression in distal para-cancerous tissues of patients with CRC (×400). (G) LDHB expression in CRC tissues (×100). (H) LDHB expression in CRC tissues (×400). (I) Immunohistochemical scores of LDHB in cancer tissues from 30 patients with CRC. (J) Correlation between immunohistochemical scores of CKMT2 and LDHB in tissues of patients with CRC. (K) Co-immunoprecipitation to detect the binding status between CKMT2 and LDHB.

Discussion

The phosphocreatine-creatine kinase (CK) shuttle system is foundational for cellular ATP homeostasis. ATP produced by glycolysis and mitochondrial oxidative phosphorylation undergoes reversible phosphate transfer with PCr under the catalysis of CK. This transfer, facilitated by the PCr-CK shuttle, maintains cellular energy homeostasis. CK is involved in the regulation of mitosis and is closely related to the cell cycle progression, cell death, and energy metabolism. CK consists of four isoforms: two are cytosolic CKs, CKM and CKB, which maintain the cytosolic ADP/ATP ratio and promote ATP consumption processes; the other two isoforms are mitochondrial CKs, CKMT1 and CKMT2, located within mitochondria. As enzymes of the creatine/phosphocreatine system, they play a prominent role in the complex metabolic energy transfer network within cells (Yan, 2016). In cancer cell proliferation and spread, mitochondrial CKs, beyond their roles in ATP generation and energy transduction, also facilitate mitochondrial respiration to meet the diverse material needs of cancer cell growth. They function through ROS and Akt pathways to regulate cell signaling transduction and phospholipid transfer (Schlattner, Kay & Tokarska-Schlattner, 2018). CKMT1 and CKMT2 are encoded by two separate nuclear genes, yet their coding sequences are highly homologous. CKMT1 is expressed in many tissues, whereas CKMT2 is primarily expressed in striated muscle tissues (Whittington et al., 2018).

Kita et al. (2023) discovered that CK regulate creatine shuttling and oxidative phosphorylation, playing a crucial role in energy metabolism and phosphorylation signaling transduction in CRC. For instance, CK inhibition resulted in decreased phosphorylation of the epidermal growth factor receptor (EGFR), protein kinase B (AKT), and extracellular regulated protein kinases 1/2 (ERK1/2). Inhibiting the creatine shuttle reduces mitochondrial respiration, lowers mitochondrial membrane potential, increases mitochondrial ROS levels, and the reduced mitochondrial respiration cannot be compensated by glycolysis (Kita et al., 2023). Though the Warburg effect is the primary energy production mechanism in tumors; the creatine shuttle is also indispensable for the aggressive proliferation of cancer cells (Yan, 2016). Additionally, the work of Kita et al. speculated that CKMT1 had a higher affinity for creatine and ATP than CKMT2; hence, no specific studies were conducted on CKMT2 (Kita et al., 2023). The role of CKMT2 in tumor cell energy metabolism requires further research.

The CKMT2 gene is situated on chromosome 5q13.3 in humans and encompasses 11 exons, with a sequence length surpassing 37 kb. CKMT2 exists in two structural forms, forming either dimers or octamers, with these two low-molecular-weight forms being capable of interconversion (Park et al., 2021). Playing a pivotal role in various biological processes, CKMT2 contributes to muscle contraction, ATP generation, ketone metabolism, intracellular energy transport, and the biosynthesis of modified amino acids. While previous studies have associated CKMT2 expression with malignancies such as osteosarcoma (Zhang et al., 2022) and gastric cancer (Ye et al., 2021), there are few reports on the biological functions and molecular mechanisms underlying CKMT2 in tumors. This study contributes evidence of abnormal CKMT2 expression in CRC, supported by both clinical tissue specimens and in vitro cell experiments, highlighting elevated CKMT2 expression in CRC. The findings suggest that CKMT2 exhibiting oncogenic roles in CRC.

In vitro cell experiments, the expression of CKMT2 showed various expression profiles in different CRC cell lines. Among them, SW480, DLD-1, HCT116, and HCT15 originated from primary lesions of human colorectal cancer, while SW620 was derived from an abdominal metastatic lesion of a colorectal cancer patient (Chen et al., 1995; Gagos et al., 1995; Nathan, Burkhart & Morin, 1990). The SW480 and SW620 cell lines were established from biopsies obtained at different times from the same patient, with the SW480 cell line derived from a primary colon carcinoma exhibiting an epithelial-like morphology in vitro, In contrast, the SW620 cell line was isolated from an abdominal metastatic lesion, displaying a fibroblast-like appearance with high tumorigenicity and metastatic potential (Gagos et al., 1995; Hewitt et al., 2000). Given that this study focuses on the biological behavior changes in primary colorectal cancer, elucidating mechanisms at the primary lesion rather than primarily investigating secondary tumors, the SW480 cell line, a representative model of primary cancer at the genomic level and widely used in experiments, was chosen for subsequent cell experiments. In addition, the DLD-1 and HCT15 cell lines, derived from the same cancer specimen (Chen et al., 1995). In this study, though the increased expression of CKMT2 in SW480/SW620 cell lines and the reduced expression in DLD-1/HCT15 cell lines were observed, the reason remained unclear. The different expression profiles of CKMT2 between SW480/SW620 and DLD-1/HCT15 might be due to genetic heterogeneity among different cells, however, further researches are warranted to investigate the detailed mechanisms underlying different expression profiles of CKMT2 in CRC cell lines.

The Warburg effect, also recognized as aerobic glycolysis, is a distinct metabolic process prevalent in cancer cell growth. It refers to the reliance of cancer cells on aerobic glycolysis instead of oxidative phosphorylation to derive energy (Fukushi et al., 2022). Compared to oxidative phosphorylation, the Warburg effect provides a quicker source of ATP, supporting rapid tumor cell division. Numerous studies have reported the pivotal role of the Warburg effect in fostering the occurrence and progression of various malignant tumors, including CRC (Johar et al., 2021; Karaca et al., 2022). Zhong et al. (2022) have suggested that the Warburg effect not only promotes CRC metastasis but also reshapes the tumor microenvironment, establishing it as a marker and therapeutic target for CRC. This study sheds light on the involvement of CKMT2 in regulating glucose metabolism, thereby promoting the Warburg effect in CRC cells.

LDH, a glycolytic enzyme composed of LDHA and LDHB subunits, plays a crucial role in the interconversion of pyruvate and lactate in glycolysis. Tumor cells, characterized by high metabolic plasticity, exhibit adaptability in substrate selection, accelerating glycolysis to generate ATP rapidly and meet cellular growth demands. LDHB, a subunit of LDH, is upregulated in various malignant tumors and is correlated with aerobic glycolysis (Urbańska & Orzechowski, 2019). A study by Wang et al. (2021) indicated that by upregulating LDHB expression, HYOU1 promotes aerobic glycolysis and malignant progression in cells of papillary thyroid carcinoma. Furthermore, LDHB has been implicated in regulating lysosomal acidification and the autophagic process, accelerating the growth of CRC cells (Shi et al., 2019). Literature has also revealed interactions between Aurora-A and LDHB, regulating LDHB phosphorylation to enhance the conversion of pyruvate to lactate, thereby promoting the Warburg effect and mediating cancer progression (Cheng et al., 2019). This study reveals that by modulating LDHB, CKMT2 reprograms glucose metabolism and enhances the Warburg effect, contributing to CRC occurrence and development. CKMT2 is a phosphate kinase. One of its mechanisms of action in cancer cells is the phosphorylation of related proteins, thereby affecting their biological functions. Through a reciprocal Co-IP assay, this study verified the association between CKMT2 and LDHB. Therefore, we speculate that CKMT2 may promote the Warburg effect by mediating the phosphorylation of LDHB. However, further investigations are warranted to delineate the detailed mechanisms underlying the interactions between CKMT2 and LDHB in promoting the Warburg effect, as well as the specific pathways through which CKMT2 mediates LDHB upregulation.

Conclusions

In summary, this study reveals that CKMT2 is highly expressed in CRC; in addition, it shows that CKMT2 could be used as a prognostic indicator linked to survival of CRC patients. In vitro studies suggest the pivotal role of CKMT2 in amplification the Warburg effect through upregulation of LDHB. This molecular mechanism significantly influences the initiation and progression of CRC.

Supplemental Information

Supplemental Information 1 Clinicopathological features for the 220 patients with colorectal cancer.

Supplemental Information 2 STROBE checklist.

Supplemental Information 3 Uncropped gels/blots.

Supplemental Information 4 MIQE checklist.

Additional Information and Declarations

Competing Interests

Author Contributions

Human Ethics

Data Availability

The authors declare that they have no competing interests.

Shasha Cai conceived and designed the experiments, analyzed the data, prepared figures and/or tables, and approved the final draft.

Qingqing Xia performed the experiments, prepared figures and/or tables, authored or reviewed drafts of the article, and approved the final draft.

Darong Duan performed the experiments, prepared figures and/or tables, and approved the final draft.

Junhui Fu performed the experiments, prepared figures and/or tables, and approved the final draft.

Zhenxing Wu analyzed the data, authored or reviewed drafts of the article, and approved the final draft.

Zaixing Yang conceived and designed the experiments, analyzed the data, authored or reviewed drafts of the article, and approved the final draft.

Changfa Yu conceived and designed the experiments, performed the experiments, prepared figures and/or tables, authored or reviewed drafts of the article, and approved the final draft.

The following information was supplied relating to ethical approvals (i.e., approving body and any reference numbers):

The ethical committee of Taizhou First People’s Hospital provided approval to carry out the study within its facilities (Ethical Application Ref: 2019-KY-009-03).

The following information was supplied regarding data availability:

Raw data are available in the Supplemental Files.

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
