# Peer review of "Creatine kinase mitochondrial 2 promotes the growth and progression of colorectal cancer via enhancing Warburg effect through lactate dehydrogenase B"

_PeerJ, doi:10.7717/peerj.17672_

## Round 0.1 · original submission · Major Revisions

The reviewer's comments on the study's rationale should be clarified, and all comments on the experimental design and findings should be addressed.

Reviewer 1 ·

Basic reporting

It was observed that there is lack of explanation in the rationale/background of the study.

The introduction about the mechanism CKMT2 in cellular process should be explained in more detail. Especially the more information is required to show and improve the rationale the relationship between lactate dehydrogenase B (LDHB) . I suggest that you improve the description at lines 57- 86 to provide more justification for your study.

It is not clear which underlying mechanisms will be focused in this study and why these mechanism are choosen?

Experimental design

Biogical and technical replicates have been never mentioned in manuscript?
In Figure 1 scale bars are missing
In siRNA and glucose uptake assays, why primary tumor derived SW480 cells with epithelioid morphology were choosen instead of metastatic SW620 despite of similar relative expression of ckmt1?
5 different CRC cell lines were used in current study. What are the differences of these cell lines from each other?

Validity of the findings

There are gaps in the explanation of rationales throughout the study. This makes it difficult to read and relate the results. For example, why was the physical interaction between LDHB and CKMT2 examined? As mentioned in the discussion it was that revealed “interactions between Aurora-A and LDHB, regulating LDHB phosphorylation to enhance the conversion of pyruvate to lactate, thereby promoting the Warburg Effect and mediating cancer progression (line 360)”. In this case, does the association of LDHB2 with CKMT2 indicate that it is phosphorylated by CKMT2?

Reviewer 2 ·

Basic reporting

There are some wording problems that make understanding difficult. For example, in the “background” part of the text and “line 207”, “heightened” is used to indicate increased gene expression. In relation to gene expression “heightened” is not a commonly used word and needs to be rephrased. At line 203 “collating” should be corrected to “Collecting”, in line 206 “comparted” should be corrected to “compared”. There may be other unnoticed drafting errors that should be traced and corrected.

This manuscript doesn’t provide enough information, which makes it difficult to understand and interpret. The points below should be taken into account when revising.

1. There should be information about the main types of cellular energy systems, including the phosphocreatine shuttle system, and how they're related to each other. How are they manipulated in cancer? The phosphocreatine shuttle system was mentioned in the text, but not explained by reference, although it is the main topic of the text. There should be information about why the phosphocreatine system is needed? why is it manipulated in some cancers? Which proteins are involved? What is the difference between CKMT1 and CKMT2 isoforms?

2.There is also a recent paper that is closely related to this study (Kita M., 2023; Oncotarget, 2023, Vol. 14, pp: 485-501) that details the creatine kinases (cytosolic creatine kinase B and CKMT1, an isoform of CKMT2) associated with colorectal cancer. In the paper by Kita M et al, the effect of CKB and CKMT1 inhibition was studied in terms of cell proliferation, stemness markers, mitochondrial functions, glycolysis, metastasis and phosphate signalling. In this paper they explain why they chose CKMT1 rather than CKMT2. They also found that inhibiting creatine kinase doesn't affect glycolysis, but does affect the phosphorylation of signalling receptors such as EGFR. Although this paper was mentioned in the manuscript (line 69), it was not properly discussed in the text.

3.They found an important link between CKMT2 and aerobic glycolysis producing lactose, known as the Warburg effect. This relationship should be discussed further. In addition, the background section should further clarify the scope of the methods used. For example, qRT-PCR was used to analyse the expression levels of CMKT2 in different colorectal cancer cell lines.

Experimental design

The experimental design is appropriate for the scope of the study. The research question is well defined. However, there are some concerns about the Materials and Methods section.

1.Concentration of RNA should be reported in line 107 instead of volume.

2.In line 110, instead of "compared" , "normalised" can be used as it is more common terminology.

3. The method of qRT-PCR, comparative or standard, should be stated.

4.The colorectal cell lines used in the study are not clearly explained. Why were these cell lines chosen? What are the differences between them?

5. In the co-immunoprecipitation experiment, line 160, the reason for using the anti-myc antibody should be clarified. The nucleotide sequence of the CKMT2 siRNA and its control should be provided

Validity of the findings

These are the concerns about the data and its interpretation:

1. Figure 1F; Among the colorectal cell lines, the significant increase of CKMT2 expression was detected in SW480 and SW620 cell lines and this result is highlighted. On the contrary, there is a significant decrease in CKMT2 levels in DLD-1, HCT15 and HCT116 cell lines compared to the control. This result, which is not reported in the manuscript, should be reported and discussed with reference in the text.

2. Figure 3B, the protein markers should be indicated on the Western blot figure. In the raw data it is not clear whether the membrane used for CKMT2 analysis was reprobed with GAPDH antibody. The protein markers should also be indicated in the raw data provided. The membranes should be labelled as the first antibody and reprobed with the second antibody for ease of understanding.

3. In Figure 4D, the si-RNA experiment data should be accompanied by Western blot analysis to ensure that the CKMT2 si-RNA works well for NCM460 control cells.

4. In Figure 5N, in addition to LDHB, there was a significant decrease in H2B, another factor involved in the glycolysis process. However, H2B was not discussed at all.

5. In Figure 50, the expression of CKMT2 needs to be checked to be sure that CKMT2 si-RNA has worked well.

6. In Figure 6A it is important to show the immunohistochemical staining of the paracancerous cells as well.

7. For Figure 60; totals or inputs of cell lysate should be included in the Western blot image.

8. The results need to be discussed further in the light of previous reports, in particular the results of Kita M et al, 2023.
9. In the Discussion section, line 337, there is a sentence suggesting a tumour suppressor role for CKMT2. However, this is not discussed.

10. In this study, CKMT2 was linked to aerobic glycolysis, but the link between aerobic glycolysis and creatine shuttle was not discussed.

Additional comments

Since cancer cells use the creatine shuttle to have more energy to proliferate and migrate, the question asked in the study is important. In addition, their results suggest a link between the creatine shuttle and aerobic glycolysis. However, the explanation, presentation, and discussion of these issues have not been properly addressed. Therefore, this manuscript needs to be substantially revised in light of these suggestions.

Reviewer 3 ·

Basic reporting

There are some grammatical errors in the text such as;
line 206 "comparted with" has to be replaced with "compared with"
line 268 "TRANSWELL" has to be corrected as "Transwell"
line 120 1640 DMEM has to be checked

All gene names have to be written in italics

Experimental design

lines 83-88 In the study design section, the design of the study can not be understood whether it is a prospective or retrospective study. The inclusion criterias have to be revised.
A general information regarding LDHB has to be included in the introduction setion.
line 100 the catalogue numbers for CKTM2 and LDHB are missing
lines 105/144/161/ which cell lines are used in these experiments?


In figure 4G, the term lactate production is not compatible with the text

Validity of the findings

no comment

Additional comments

The expression of CKMT2 in all of the cell lines have to be discussed.
Although there was an increase in CKMT2 expression both in SV480 and SV620 cell lines, why only the SV480 results have been discussed.

---

## Round 0.2 · Major Revisions

Although the authors answered several reviewers' comments, revisions are still needed.

Reviewer 1 ·

Basic reporting

The authors responded to my comments.Issues were added and discussed. I have no additional comments

Experimental design

The failed and missing parts of the experimental design were added in the methods and results parts.

Validity of the findings

no comment

Reviewer 2 ·

Basic reporting

In general, the language is not clear and unambiguous. There are statements which are not supported by references. There are statements in the manuscript that are not self-contained with relevant results to hypothesis.
The answers to the comments, which were previously stated, are considered. The statements that need revision are indicated; others should be considered as accepted.

8. Comment:
There should be information about the main types of cellular energy systems, including the phosphocreatine shuttle system, and how they're related to each other. How are they manipulated in cancer? The phosphocreatine shuttle system was mentioned in the text, but not explained by reference, although it is the main topic of the text. There should be information about why the phosphocreatine system is needed? Why is it manipulated in some cancers? Which proteins are involved? What is the difference between CKMT1 and CKMT2 isoforms?

Issues about revision: Line 391-406. The information about types of cellular energy systems is added to the discussion part of the manuscript. At line 396 -------CK consists of four subunits----instead of subunit “isoform” should be used. At line 402 the kind of respiration should be indicated. At line 406 the reference is missing.

9. and 23. comments:
There is also a recent paper that is closely related to this study (Kita M., 2023; Oncotarget, 2023, Vol. 14, pp: 485-501) that details the creatine kinases (cytosolic creatine kinase B and CKMT1, an isoform of CKMT2) associated with colorectal cancer. In the paper by Kita M et al, the effect of CKB and CKMT1 inhibition was studied in terms of cell proliferation, stemness markers, mitochondrial functions, glycolysis, metastasis and phosphate signaling. In this paper they explain why they chose CKMT1 rather than CKMT2. They also found that inhibiting creatine kinase doesn't affect glycolysis, but does affect the phosphorylation of signaling receptors such as EGFR. Although this paper was mentioned in the manuscript (line 69), it was not properly discussed in the text.

Issues about the revision ( Line 407-419) Kita M. 2023 paper is not properly interpreted. In this paper they analysed the importance of creatine kinases by using creatine kinase inhibitors that effect all types of creatine kinases including cytoplasmic and mitochondrial. Therefore, it will not be appropriate to emphasize CKMT1 and CKB and exclude CKMT2. Usage of CK inhibitors blocks signaling pathways including EGFR,AKT and ERK1/2. Therefore, it is not necessary to emphasize EGFR. Moreover, both MTCK and CKB knock-down resulted in decreased EGFR phosphorylation. Therefore, emphasis on CKB might be misleading. At Lines 412-416, the statements are given without reference. These statements are not indicated in Kita et al 2023 paper. In Kita et al paper, they showed that creatine kinase inhibitors cause reduced mitochondrial respiration which is not compensated by glycolysis and lactate fermentation.In addition, at line 417 there is a statement saying that “ Additional, this study found that CKMT1 has a higher affinity……..” which refers to Kita et al study. Actually, Kita et al did not find it. Based on this information in Kita et al preferred to study CKMT1 instead of CKMT2. In general this paper need to be evaluated correctly.

10.comment
They found an important link between CKMT2 and aerobic glycolysis producing lactose, known as the Warburg effect. This relationship should be discussed further. In addition, the background section should further clarify the scope of the methods used. For example, qRT-PCR was used to analyse the expression levels of CMKT2 in different colorectal cancer cell lines.

Issues about revision (Lines 96-106): First the relation between CK and glycolysis. Then CKMT2 and glycolysis is explained without connection between sentences. Is there an information about CKMT1 and glycolysis in the literature? If there is it should be included as well. Concerning the line 106-113, by mistake previously I made this comment “In addition, the background section should further clarify the scope of the methods used. For example, qRT-PCR was used to analyse the expression levels of CMKT2 in different colorectal cancer cell lines.” Instead of background section the scope of eperiments should be integrated into methods section of abstract. In addition, the end of the interaction should be the punchline sentence not the information about methods.

Experimental design

These are the concerns about the experimental approaches.

22. Comment.
For Figure 60; totals or inputs of cell lysate should be included in the Western blot image.
Issues about revision Figure 6K. Input is the starting material for Immunoprecipitation (IP) experiment. Input is checked for protein of interests to normalize the obtained result from IP. In normal, IP for IgG and CKMT2 or LDHB antibodies containing samples should start with the same amount of protein lysate. However, at figure 6K, it looks like there is no protein in IgG antibody containing sample. Moreover, in line 388 there is a statement such “Thus, CKMT2 and LDHB may interact directly in CRC cells” which is not appropriate because to analyse direct interaction only the purified proteins, CKMT2 and LDHB, should be tested in the absence of other proteins in the lysate.

Validity of the findings

Conclusions are not well stated as indicated below.

16. Comment about the selection of cell lines and the interpretation of results.
Figure 1F; Among the colorectal cell lines, the significant increase of CKMT2 expression was detected in SW480 and SW620 cell lines and this result is highlighted. On the contrary, there is a significant decrease in CKMT2 levels in DLD-1, HCT15 and HCT116 cell lines compared to the control. This result, which is not reported in the manuscript, should be reported and discussed with reference in the text.

Issues about revision (Line 254-259) The statements do not support the results. For instance, increase in CKMT2 expressions detected in SW480 and SW620. These cell lines derived from the same individual. The difference between them is their metastatic ability. SW480 is primary colon carcinoma. SW620 is metastatic. So these cells lines are from different stages of cancer, but both of them show significantly high CKMT2 expression compared to control and other primary lesions of colorectal cancer. DLD1 and HCT115 cells derived from the same cancer specimen, both of them showed low expression profile of CKMT2 expression compared to control and this is correlated with unrelated karyotypes. These explanations don’t make any sense. Here, SW480 and SW620, DLD-1 and HCT15 show common expression profiles, one group increased expression and the other group reduced expression. The question is what is the difference between SW480/SW620 and DLD-1/HCT15. The knowledge about the cell lines were given, however it is not interpreted in terms of obtained results. In addition, the discussion of the results should be at discussion part.

Additional comments

Upon this revision the manuscript is improved. However, there are still serious issues about the interpretation of the previous literature and discussion of the results. The contingeny is missing.

Therefore, this manuscript still needs to be substantially revised in light of these suggestions.

---

## Round 0.3 · accepted · Accept

Please consider the reviewer's final suggestions; the manuscript will be ready for publication.

Reviewer 2 ·

Basic reporting

These are my evaluations for the manuscript that has been previously asked for two major revisions. To make the understanding easier, I kept the order and number of comments the same.
8. Comment:
The comments are considered and revised accordingly. (Line 398,404,408-409)
9 and 23.Comments:
At line 412 -phosphorylation signaling transduction------is not appropriate terminology, it needs rephrasing.
10.Comments:
The information about the link between CKMT1 and glycolysis has been added to the text (lines 106-108). However, what the GC abbreviation stands for was not indicated.

Experimental design

Comment 10: and 22
The revision about experimental issues were considered and corrected, Line 35-39 and Figure 6K

Validity of the findings

16.Comment
The language should be improved in general. In particular, Line 439: ‘In vitro cell experiments, the expression of CKMT2 showed various expression profiles in different CRC cell lines.’sentence needs paraphrasing and correction.
‘primary cancer at the genomic level’ statement (line 450) should be explained. It is not common terminology.